# Application of flexible parametric cure model in determination of effective factors on the technique of survival peritoneal dialysis patients in adults of Iran

Maryam Karimi Ghahfarokhi[1,2], Mehdi Yaseri[2]*, Mostafa Hosseini[2]*, Fatemeh Masaebi[3]

**1** Department of Biostatistics, School of Allied Medical Sciences, Shahid Beheshti University of Medical Sciences, Tehran, Iran, **2** Department of Epidemiology and Biostatistics, School of Public Health, Tehran University of Medical Sciences, Tehran, Iran, **3** Department of Biostatistics and Epidemiology, School of Medicine, Zanjan university of medical sciences, Zanjan, Iran

* m.yaseri@gmail.com (MY); mhossin110@yahoo.com (MH)

## Abstract

### Objective

Peritoneal Dialysis (PD) is one of the most common methods of dialysis performed in patients with kidney failure. This study presents the application of flexible parametric cure models, which enhance fit and interpretability by accommodating various survival distributions and allowing for the estimation of the cured fraction in diverse populations. Thus, the purpose of this study is to determine the risk and demographic factors affecting the survival of non-cured dialysis patients undergoing peritoneal dialysis, as well as to estimate the cured fraction of survival using this method.

### Methodology

A total of 4,144 adult patients (aged 20 and above) were included in a retrospective (historical) cohort study from the Iranian PD registry, conducted between 1995 and 2018 across 20 centers nationwide. The study considered important variables such as sex, age, BMI, creatinine, calcium, cholesterol, triglycerides, Low-Density Lipoprotein (LDL), High-Density Lipoprotein (HDL), Erythrocyte Sedimentation Rate (ESR), systolic blood pressure (SBP), and diastolic blood pressure (DBP). A flexible parametric cure model was employed to assess the independent risk factors for PD technique failure.

### Findings

The results of the current study showed that the flexible parametric cure Model (AIC = 5735.06) has a better fit compared with another cure model (mixture cure model = 6676.25, non-mixture cure model = 6677.28). Based on the findings of this

**Data availability statement:** Data cannot be shared publicly because it is a national information registration system of peritoneal dialysis patients in Iran. Data are available from the Tehran University of Medical Sciences Institutional Data Access / Ethics Committee (contact via Ethics Committee of Tehran University of Medical Sciences Email: Ethics@sina.tums.ac.ir) for researchers who meet the criteria for access to confidential data. We obtained data from the Shefa Peritoneal Dialysis Center in Iran. All peritoneal dialysis patients with complete records from 20 dialysis centers across the country are investigated, along with data from ten registered questionnaires. The questionnaires were sent from all over the country to a Shefa Peritoneal Dialysis Center in Tehran and entered into comprehensive software named "Hakim" (a Persian-language software for recording medical records of patients in hospitals and healthcare service centers). The necessary information for this research was extracted from the Jame Hakim software. We can provide the existing data file on which this article is based.

**Funding:** The author(s) received no specific funding for this work.

**Competing interests:** The authors have declared that no competing interests exist.

study, the estimated cure fraction of survival using the PD method was 50% (95% CI: 0.47–0.53). The flexible cure model, demonstrated in a multivariate setting, indicated that variables such as age over 60 years (HR = 2.11, 95% CI: 1.05–1.35), calcium levels above 10.2 (HR = 1.43, 95% CI: 1.30–1.56), LDL levels above 140 (HR = 1.23 95% CI: 1.07–1.42), ESR (HR = 1.006, 95% CI: 1.003–1.01) and DBP (HR = 1.17, 95% CI: 1.16–1.31) significantly impacted the survival of non-cured patients.

## Conclusion

The presence of cured individuals in the data flattens the Kaplan-Meier curve. In such cases, employing a flexible parametric cure model is appropriate to separately investigate the effects of various factors on the cure fraction and the survival of susceptible individuals. The findings of this study indicated that age, calcium, LDL, ESR, and DBP significantly influenced the survival of non-cured individuals. Specifically, increases in these variables were associated with a higher risk of technique failure.

## Introduction

Peritoneal Dialysis (PD) is a widely utilized renal replacement therapy that supports kidney function in patients with end-stage renal disease (ESRD) but does not cure the underlying condition [1]. PD is available in two forms: continuous ambulatory peritoneal dialysis (CAPD) and automated peritoneal dialysis (APD), with only CAPD being performed in Iran. As PD is widely recognized as an effective treatment for kidney failure, extensive information has been gathered on its efficacy and its role in reducing complications [2]. Its popularity has increased due to its simplicity, easy access, and relatively affordable cost [3]. However, the utilization rate of PD varies among different countries, ranging from zero to over 60% of the dialysis population, influenced by socioeconomic conditions, healthcare systems, and patient support factors [4].

PD is considered unsuccessful when it no longer effectively manages kidney function replacement, necessitating a transition to hemodialysis or kidney transplantation. Causes of PDfailure include peritonitis, ultrafiltration failure, kidney transplantation, and even cultural and social challenges that can develop over time in many cases [5]. PD failure is a valuable indicator for assessing the condition of patients undergoing this treatment; however, the associated complications are complex and can potentially lead to patient mortality. Therefore, it is important to explore alternatives that can predict the impending failure process before it occurs, one of which is the use of statistical methods.

Historically, studies analyzing CAPD patient survival have predominantly relied on conventional statistical models, such as the log-rank test and the Cox proportional hazards model [6–8]. However, these models assume that all individuals in the study are at risk of experiencing the event (death) or being censored. This assumption does not align with PD, as it is a maintenance treatment where a substantial proportion of patients do not experience death during follow-up. For example, while Kaplan-Meier

survival estimates provide useful insights, they fail to account for patients who are effectively "cured" of the event of interest, leading to potential biases. Similarly, the Cox proportional hazards model assumes that all individuals remain susceptible to the event throughout the study period, which does not adequately capture the heterogeneity in PD populations.

A more appropriate approach involves cure models [9–14], which divide the population into two groups: susceptible individuals and those considered cured or immune from the event. Cure models assume that a subset of patients will never experience the event, resulting in a plateau in the survival curve. These models estimate both the proportion of cured individuals and the survival function of those who remain susceptible, while also examining the factors influencing both outcomes [15]. Cure models are broadly classified into two types: mixture models and non-mixture models [16]. The reliability of these models depends on various factors, including the duration of follow-up and the selection of an appropriate survival distribution (e.g., exponential, Weibull, or logistic) for susceptible individuals [17]. One of the key challenges in survival analysis is ensuring model flexibility to accurately capture the distribution of event times [18,19].

To address these limitations, this study employs a flexible parametric cure model, a variant of the non-mixture cure model, which allows for greater adaptability in modeling survival outcomes [17]. To the best of our knowledge, this is the first study to apply a flexible parametric cure model to assess PD technique failure among all adult PD patients (aged 20 and older) in Iran. The study leverages data from the national computerized PD registry system to determine the technique failure rate and its associated risk factors. By incorporating a robust statistical framework, this research aims to provide a more accurate and comprehensive understanding of PD failure, ultimately guiding clinical decision-making and improving patient management strategies.

## Materials and methods

Baseline characteristics and laboratory data of 4,144 Iranian patients (younger than 20 years) were included in a retrospective (historical) cohort study based on the Iranian PD registry, which has continuously recorded data from 1995 to February 2018 across 20 centers nationwide. These data were extracted from the registry and entered into Hakim software (Electronic Health Record: Pegahsoft, Tehran, Iran). The demographic and clinical characteristics of PD patients in the country are documented by specialists using ten questionnaires, which are then entered into this system. Hakim software serves as a national database that records information about PD patients. Details regarding the registry and data collection have been reported in previous studies [20,21].

The Ethics Committee of Tehran University of Medical Sciences approved the study protocol (approval ID: IR.TUMS.SPH.REC.1399.314). After cleaning, identifying, and addressing potential errors in the record entries, the data were finalized for analysis. Complete records were extracted and utilized in this study.

The effect of variables such as sex, age, appetite, BMI, blood hemoglobin, creatinine, phosphate, fasting blood glucose, calcium, cholesterol, triglycerides, LDL, HDL, albumin, parathyroid hormone, ESR, SBP, DBP, GFR, and total body iron binding capacity on patient survival was assessed. Standard ranges were used for the classification of quantitative variables [22,23].

Any death due to PD was considered a failure, and patients who received a kidney transplant or dropped out of the study were classified as censored. Specifically, any patient who died while on PD was classified as having experienced the event of interest, which aligns with the clinical objective of evaluating survival outcomes in PD patients, where mortality serves as the ultimate endpoint. Similarly, patients who transitioned to hemodialysis or were placed on the kidney transplant emergency list were also considered censored. This decision was made to ensure that individuals who switched to alternative treatment modalities, such as hemodialysis or kidney transplantation, were not mistakenly categorized as having failed PD. In this context, censoring is essential to prevent misclassification, as these patients continued to receive care, albeit through different treatment options, and no longer relied on PD for renal replacement therapy.

This study employs a flexible parametric survival model to estimate the survival rate of PD in adults in Iran and investigate the influential factors on the long-term survival (stability) of PD in patients with kidney failure. The flexible parametric survival model is fitted using STATA version 14 software and the subprogram stpm2.

## Statistical analysis

### Model diagnostics

Before using the cure model, the assumptions of the model were carefully examined. These assumptions include the presence of patients with long-term survival and sufficient duration of the study period, which were tested using statistical tests.

At first, the presence of immune individuals was first visually assessed using Kaplan-Meier estimates. In datasets with immune individuals, censored observations are typically observed at longer time points on the right side of the Kaplan-Meier survival curve. A flattening of the curve at a value greater than zero suggests the presence of both susceptible and immune individuals.

Statistical tests were used then to examine the proportion of individuals whose censoring time exceeds the largest event occurrence time. The question arises as to whether a significant percentage of individuals have long-term survival or not. In some cases, clinical experiences and biological evidence can demonstrate that a proportion of cured individuals exists. Otherwise, using statistical tests can be helpful. This hypothesis can be written as $H_0 : p = 1$, meaning that under the assumption , $H_0$, all individuals are at risk of the event occurring, and there is no proportion of cured individuals. In the proposed method by Maller and Zhou, a test based on the i.i.d censoring model with the assumption of exponential and uniform censoring distribution is presented. Critical values for accepting or rejecting the existence of a proportion of cured individuals have been compiled in a table. This table is based on the censoring model and the uniform and exponential censoring distribution. To use this table, the estimated number of susceptible individuals is first calculated from the sample, and based on that, the appropriate censoring distribution is computed. Then, the critical value is extracted from the table, considering the sample size at a specified significance level, and based on that, the hypothesis of the existence of patients with long-term survival is accepted or rejected [24].

After obtaining sufficient evidence regarding the existence of patients with long-term survival, the second point is to ensure that the significance of the proportion of cured individuals and its difference from zero is not due to the short follow-up duration. For this purpose, Maller and Zhou have developed tables under the conditions of the previous test (i.i.d censoring times and uniform or exponential censoring distribution) through simulation. The following steps are carried out:

1- Calculate the last observed failure time $t^*_{(n)}$

2- Calculate the last observed censoring time $t_n$

3- Compute $\delta_n = t_n - t^*_{(n)}$

4- Count the number of failures in the interval $N_n - (2t^*_n - t_n, \ t^*_{(n)}]$

5- Calculate $q_n = \frac{N_n}{n}$

The obtained value of $q_n$ is compared with the critical value related to the sample size of the study and B (U[0,B]) or $\mu$ (Exp($\mu$)) from the simulation tables of Maller and Zhou [25].

### Model validation

After establishing the model assumptions, univariate flexible cure models were employed for the preliminary identification of important risk factors for technique failure. Subsequently, a backward variable selection procedure was applied in a multivariate flexible cure model (with a P-value<0.2 for elimination) to identify potential baseline risk factors affecting technique failure after adjusting for possible confounders. A P-value<0.05 was used to determine the significance of the final flexible cure model. To validate the model, the Akaike Information Criterion (AIC) was used to compare competing

models and select the one with the best balance of goodness-of-fit and complexity. The model with the lowest AIC value was chosen as the final model.

## Results

In the present study, 4144 patients were enrolled. The patients were in the age range of 22–85 years, with a mean of 52.1($\pm16.12$) years, of which 2049 (50.06%) were male. The descriptive statistics for the laboratory variables are presented in Table 1.

Based on the event of interest (until the end of February 2018), 1160 (28.34%) individuals under study experienced the event, and the remaining 2933 (71.66%) were censored. A Kaplan-Meier survival curve, with 95% confidence bands indicating the variability in the survival estimates, was plotted to examine patients with long-term survival and sufficient follow-up duration, as shown in Fig. 1. The median survival time of the patients was 9.98 years. Additionally, the 3-year, 5-year, and 10-year survival rates of the patients under study, based on the Kaplan-Meier plot (Fig. 1), were 77%, 64%, and 49%, respectively.

The graph flattens after approximately 11 years and remains level for the remainder of the study (around 7 years). The test for the existence of patients with long-term survival, under the assumption of the i.i.d. censoring model, was conducted using the Mallor and Zhou method. To investigate the presence of censored individuals, $\hat{p}_n$, which in the Mallor and Zhou method is equal to one minus the Kaplan-Meier estimator, was 0.490 in this study (corresponding to 51% long-term survival). Since there were 2,933 uncensored deaths out of 4,093 total individuals, the percentage of censoring was calculated as 71.66%, which corresponds approximately to a $\mu = 1$ exponential distribution or B = 2 uniform distribution. Considering the critical values from the Mallor and Zhou test table for a sample size of 1,000 at a 5% significance level, the critical value for the exponential distribution is 0.940, and for the uniform distribution, it is 0.960. For sample sizes larger than 1,000, simulation results suggest that this value approaches or equals 1. Since 0.490 is less than 1, the test is significant at the 0.010 level, indicating a deficit of patients with long-term survival.

Additionally, based on the study data and using the Mallor and Zhou test for the sufficiency of follow-up duration, the following values were obtained:

- $t^*_{(n)}$ = 16.78
- $t^*_{(n)}$ = 11.72
- $q_n$ = 0.102

Since $q_n$ is greater than the critical value from the test table for follow-up sufficiency, the results indicate that the follow-up duration was sufficient."

Furthermore, the model fitting plot and its comparison with the Kaplan-Meier curve and other cure models (mixture and non-mixture cure models) can be observed in Fig. 2. The flexible cure model demonstrates the best fit to the dialysis patient data. In addition to visual comparison of the models, AIC and BIC criteria were calculated for all three models when no variables were present in the model. As seen in Table 2, the flexible cure model has lower AIC and BIC compared to the other two cure models.

In order to investigate the estimation of the PD survival rate and the factors affecting its failure (method stability), the variables are entered into the survival model individually. The results of fitting the flexible survival model with the presence of predictive variables in a single-variable manner are shown in Table 3. Additionally, the cure ratio for each variable is reported in this table.

### Investigating the significance of variables in multivariate flexible cure model

In the previous section, the significance of the variables on the survival of PD was examined individually. In this part, the significance of the variables is evaluated in a multivariate manner. First, all the variables were entered into the flexible

**Table 1. General characteristics of the sample by observed survival and dead patients.**

| Variable | Category | Observed survival patients (censored + cured) | Dead patients |
|---|---|---|---|
| | | No. % | No. % |
| **Gender** | Male | 1485 (72.47) | 564 (27.53) |
| | Female | 1448 (70.84) | 596 (29.16) |
| **Age (year)** | 20–60 | 962 (89.24) | 116 (10.76) |
| | 40–60 | 1021 (74.50) | 411 (25.50) |
| | over 60 | 770 (54.88) | 633 (45.12) |
| **Appetite** | Low | 284 (64.40) | 157 (35.60) |
| | Medium | 2520 (75.54) | 954 (27.46) |
| | High | 129 (72.47) | 49 (27.53) |
| **BMI (kg/m²)** | Underweight (<18.5) | 374 (76.17) | 117 (23.83) |
| | Normal (18.5–25) | 1321 (71.41) | 529 (28.59) |
| | Overweight (> 25) | 1238 (70.66) | 514 (29.34) |
| **Hemoglobin (g/dl)** | <=11 | 2003 (71.13) | 813 (28.87) |
| | >11 | 930 (72.83) | 347 (27.17) |
| **Creatinine (mg/dl)** | <=7 | 1671 (68.79) | 758 (31.21) |
| | >7 | 1262 (75.84) | 402 (24.16) |
| **Phosphate (mg/dl)** | <3.5 | 196 (60.68) | 127 (39.32) |
| | 3.5-5.5 | 1752 (71.28) | 706 (28.72) |
| | >5.5 | 985 (75.08) | 327 (24.92) |
| **Fasting blood glucose (mg/dl)** | <=126 | 2022 (76.27) | 629 (23.73) |
| | >126 | 911 (63.18) | 531 (36.82) |
| **Calcium (mg/dl)** | <8.2 | 560 (73.68) | 200 (26.32) |
| | 8.2-10.2 | 2224 (71.49) | 887 (28.51) |
| | >10.2 | 149 (67.12) | 73 (32.88) |
| **Cholesterol (mg/dl)** | <200 | 2131 (72.02) | 828 (27.98) |
| | 200-300 | 748 (71.17) | 303 (28.83) |
| | >300 | 54 (65.06) | 29 (34.94) |
| **Triglyceride (mg/dl)** | <200 | 2286 (71.28) | 897 (28.18) |
| | 200-500 | 623 (71.44) | 249 (28.56) |
| | >500 | 24 (63.16) | 14 (36.84) |
| **LDL** | <100 | 1179 (71.72) | 465 (28.28) |
| | 100-140 | 1358 (72.00) | 528 (28.00) |
| | >140 | 396 (70.34) | 167 (29.66) |
| **HDL** | <=40 | 1105 (71.89) | 432 (28.11) |
| | >40 | 1828 (71.52) | 728 (28.48) |
| **Albumin (g/dl)** | <=3.5 | 719 (66.45) | 363 (33.55) |
| | >3.5 | 2214 (73.53) | 797 (26.47) |
| **Parathyroid hormone (pg/ml)** | <=200 | 1556 (67.77) | 740 (32.23) |
| | >200 | 1377 (76.63) | 420 (23.37) |
| **ESR (mm/hr)** | <=20 | 164 (76.28) | 51 (23.72) |
| | >20 | 2796 (71.40) | 1109 (28.60) |
| **SBP (mmHg)** | <=120 | 883 (73.10) | 325 (26.90) |
| | >120 | 2050 (71.06) | 835 (28.94) |
| **DBP (mmHg)** | <=80 | 2011 (71.80) | 790 (28.20) |
| | >80 | 922 (71.36) | 370 (28.64) |

*(Continued)*

**Table 1.** (Continued)

| Variable | Category | Observed survival patients (censored + cured) | Dead patients |
|---|---|---|---|
| | | No. % | No. % |
| **Glomerular filtration rate (ml/min)** | >2 | 1770 (72.24) | 680 (27.76) |
| | <=2 | 1163 (70.79) | 480 (29.21) |
| **Total body iron binding capacity (mcg/dl)** | <=50 | 569 (73.42) | 206 (26.58) |
| | >50 | 2364 (71.25) | 954 (28.75) |

*Survived patients: Includes cured and censored patients who cannot be distinguished without a suitable model.

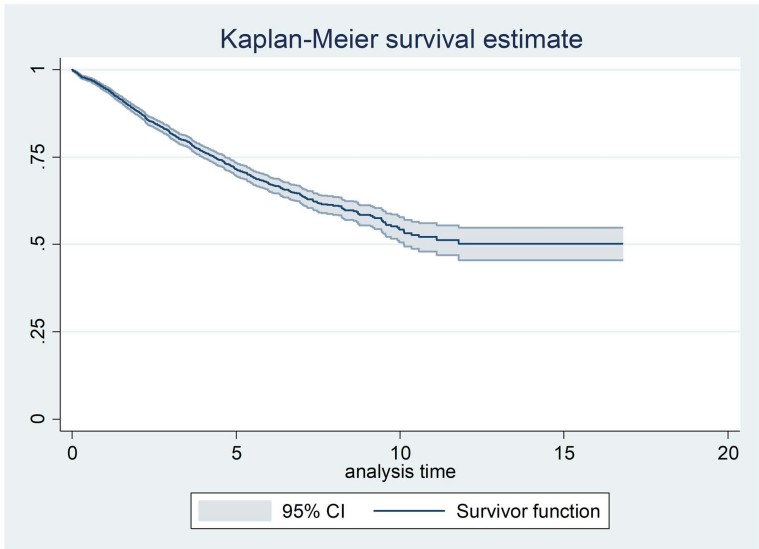

**Fig 1. Kaplan-Meier estimated survival plot based on years.** Initially, fitting a model without the presence of predictive variables was used to estimate the overall recovery rate. The results of fitting a flexible parametric cure model without predictive variables showed that the cure ratio is 0.50 (95% CI: 0.47–0.53).

survival model, and then, using backward elimination, the variables with the largest P-value above 0.1 were removed one by one from the model. Since the variables of age, LDL, and calcium showed better fit when categorized, they were entered into the model in a categorized form.

In Table 4, the final model is reported, including the model coefficients, standard errors of the coefficients, Hazard Ratios, significance of the coefficients, 95% confidence intervals, and the estimated cure ratio for each variable.

Ultimately, the variables of age, calcium, LDL, ESR, and DBP showed a significant effect on the survival of non-cured patients.

As an example, the risk in the older age group (60 years and above) is 20% higher than the risk associated with the failure of the PD method for individuals under 40 years old ($\exp(0.188) = 1.2$). In the 40-to-60-year age group, the risk is 4% higher than in the uunder-40 group, which is not statistically significant. Additionally, the recovery fraction in the age group 60 years and above ($\exp(-\exp(0.188)) = 0.3$) is lower than in the 40–60 years group ($\exp(-\exp(0.043)) = 0.35$).

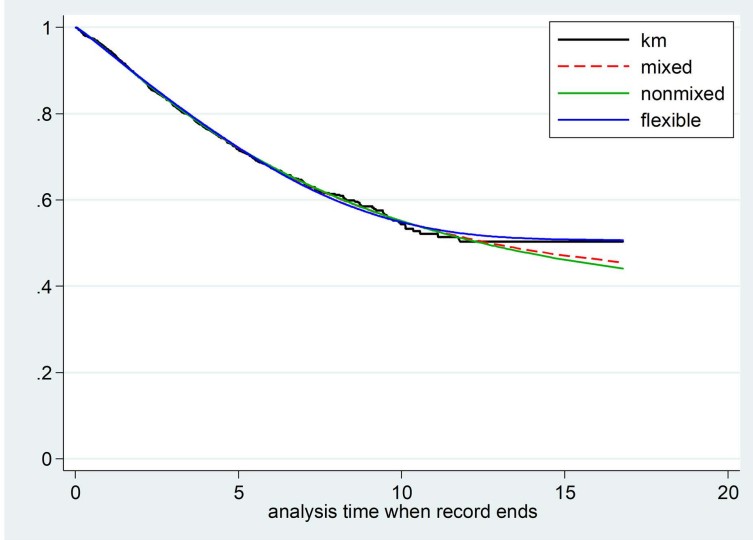

**Fig 2. Comparison of estimated survival by mixture cure models, non-mixture cure models with Weibull distribution, and flexible parametric cure model against Kaplan-Meier estimated survival.**

**Table 2. AIC and BIC criteria in selecting the model without independent variables.**

| Model | df | AIC | BIC |
| --- | --- | --- | --- |
| Mixture cure model | 3 | 6676.25 | 6695.24 |
| Non-mixture cure model | 3 | 6677.28 | 6696.27 |
| Flexible cure model | 3 | 5735.06 | 5754.05 |

## Discussion

Since the 1980s, PD has been widely used as the second kidney replacement therapy after hemodialysis. CAPD has become the most common form of PD due to its practicality, despite certain limitations. This study aimed to evaluate the technique failure rate of PD in Iranian adults and identify associated risk factors using the national PD data registry [26]. Unlike previous studies that often considered death as the primary event, this study focused on PD method failure, utilizing a flexible parametric cure model to analyze survival data. This model allowed for high flexibility in the hazard function and identified calcium level, LDL, ESR, and DBP as significant factors affecting the survival of untreated patients.

Cure models have gained significant recognition in medical science and are currently utilized by numerous researchers worldwide across various medical fields, particularly in the context of cancer patient survival [27–32]. In most studies conducted to identify the factors affecting the survival of dialysis patients, particularly those undergoing CAPD, rank test methods and the Cox proportional hazards model have been used [23,33–37]. For example, based on a cohort study, Lo et al. found that the two-year and five-year survival rates of PD patients were 88.7% and 66.7%, respectively [34].

By enhancing treatment methods and improving the health and quality of life for patients, their life experiences can align more closely with those of the general population. Consequently, cure models can serve as a valuable alternative to standard Cox proportional hazards models for analyzing survival data for several reasons. First, the assumption of proportional hazards may not hold true when survival curves exhibit plateaus at their tails. Second, survival plots with extended plateaus can reveal heterogeneity within a patient population, which is crucial to describe explicitly. Cure models allow

**Table 3. The results of fitting the flexible cure model for univariate models.**

| variable | Categories | coefficient | P-value | HR | Confidence interval for HR | |
|---|---|---|---|---|---|---|
| | | | | | lower | upper |
| gender | Women | | | | | |
| | men | 0.005 | 0.938 | 1.005 | 0.869 | 1.389 |
| age | | 0.006 | 0.024 | 1.006 | 1.001 | 1.011 |
| appetite | No appetite | | | | | |
| | Moderate appetite | 0.120 | 0.525 | 1.127 | 0.625 | 1.271 |
| | High appetite | 0.130 | 0.220 | 1.139 | 0.719 | 1.080 |
| BMI | | 0.007 | 0.260 | 1.007 | 0.995 | 1.019 |
| Hemoglobin | | 0.043 | 0.016 | 1.044 | 0.411 | 0.992 |
| Creatinine | | 0.024 | 0.004 | 1.024 | 1.002 | 1.046 |
| Phosphate | | 0.038 | 0.513 | 1.039 | 0.928 | 1.162 |
| Fasting blood glucose | | 0.0005 | 0.326 | 1.000 | 0.998 | 1.021 |
| Calcium | | 0.400 | <0.01 | 1.491 | 1.271 | 1.733 |
| Cholesterol | | 0.004 | 0.956 | 1.004 | 0.874 | 1.137 |
| triglyceride | | 0.0005 | 0.177 | 1.001 | 0.999 | 1.001 |
| LDL | | 0.002 | 0.095 | 1.002 | 0.998 | 1.003 |
| HDL | | 0.027 | 0.690 | 1.027 | 0.582 | 1.116 |
| Albumin | | 0.130 | 0.075 | 1.139 | 0.987 | 1.309 |
| parathyroid hormone | | 0.0003 | 0.091 | 1.000 | 0.999 | 1.001 |
| ESR | | 0.006 | <0.01 | 1.006 | 1.003 | 1.009 |
| SBP | | 0.004 | 0.010 | 1.004 | 1.001 | 1.007 |
| DBP | | 0.008 | <0.01 | 1.008 | 1.002 | 1.013 |
| Glomerular filtration rate | | −0.030 | 0.027 | 0.97 | 0.945 | 0.997 |
| Total body iron binding capacity | | −0.001 | 0.060 | 0.999 | 0.998 | 1.000 |

**Table 4. Results obtained from fitting the flexible parametric cure model multivariable model against the predictor variables.**

| variable | Categories | coefficient | p-value | HR | Confidence interval for HR | | Cure ratio |
|---|---|---|---|---|---|---|---|
| | | | | | lower | upper | |
| age | Age reference group | | | | | | |
| | Age 40–60 years | 0.043 | 0.605 | 1.044 | 0.912 | 1.041 | 0.35 |
| | Age over 60 years | 0.188 | 0.035 | 1.207 | 1.053 | 1.350 | 0.30 |
| Calcium | Calcium reference group | | | | | | |
| | 8.2 to 10.2 | 0.299 | 0.065 | 1.349 | 0.974 | 1.731 | 0.26 |
| | >10.2 | 0.358 | <0.001 | 1.430 | 1.321 | 1.560 | 0.24 |
| LDL | LDL reference group | | | | | | |
| | 100–140 | 0.089 | 0.405 | 1.093 | 0.893 | 1.352 | 0.40 |
| | >140 | 0.209 | 0.005 | 1.232 | 1.074 | 1.420 | 0.35 |
| ESR | | 0.006 | <0.001 | 1.006 | 1.003 | 1.010 | 0.36 |
| SBP | | 0.160 | 0.038 | 1.174 | 1.311 | 1.162 | 0.31 |

us to explore covariates associated with both short-term and long-term effects. For instance, they enable us to assess whether a new therapy influences the probability of becoming a long-term survivor or impacts survival outcomes for those who do not achieve long-term survival [38].

In a study conducted in Iran on cured models and survival analysis of patients undergoing CAPD, researchers affiliated with Akhlaghi and colleagues employed a mixture-cured model to investigate the factors influencing both long-term and short-term survival of PD patients. Their rationale for using these models was the high proportion of cured individuals (censoring) at the end of the study, given that PD is a maintenance treatment and a significant percentage of patients do not experience mortality. In this study, they applied mixture-cured survival models, including Weibull, gamma, log-normal, and logistic regression, to analyze the survival of CAPD patients. The results indicated that the Weibull model provided the best fit for most variables. Specifically, HDL, FBS, and calcium had significant effects on short-term survival, while LDL and triglycerides were on the borderline of significance. Additionally, age and albumin were significant predictors of long-term survival, with triglycerides also approaching significance. In the present study, apart from albumin and FBS, the variables calcium and LDL aligned with these findings. However, age exhibited a protective effect, which contrasts with the results of our study [39].

Escobar and colleagues conducted a cohort study in Spain from 2007 to 2013 using Cox regression and concluded that age and gender had a significant effect on PD outcomes. Notably, the age range in their study was between 20 and 75 years. In the present study, a similarly significant effect of age was observed; however, in contrast to their findings, the effect of gender was not significant [40].

Kalantar-Zadeh and colleagues, in a cohort study conducted in Iran involving 40,933 ESRD patients over 15 years, reported that the relative risk for patients with SBP above 110 mmHg was 1.6, while for hemodialysis patients with DBP below 50 mmHg, it was 2. In contrast to their findings, our model, applied to CKD patients undergoing PD, indicated that DBP has a significant effect on the survival of non-cured patients [41].

In a study by Al-company on 200 stable hemodialysis patients (without any systemic disease) and 50 hemodialysis patients with acute disease, it was shown that dialysis patients tended to have increased ESR, and an ESR > 100 is one of the causes of kidney failure. This result is consistent with the present study [42].

Advances in treatments for many chronic diseases have led to increased longevity and improved recovery in patients. Since PD is a maintenance therapy, a significant percentage of patients in the present study do not experience treatment failure. This has increased the tendency to use cured models.

The cured model is a statistical approach whose optimal properties have been studied extensively, allowing it to better capture the impact patterns of variables on outcomes over time compared to other models. The application of these models, particularly the flexible parametric cure model, has been explored to some extent in data related to patients in Iran. A better understanding of how predictive factors influence patient outcomes and the timing of their effects can enhance decision-making in the healthcare management of dialysis patients. This is especially relevant in areas such as quality of life, where such models may offer significant advantages over traditional statistical methods.

Royston and Parmar [19] proposed a flexible parametric survival model that fits the cumulative hazard on the log scale. This model directly estimates the baseline cumulative hazard using restricted cubic splines. Compared to standard parametric models, it provides greater flexibility in modeling the hazard function through the use of restricted cubic splines. This adaptability is one of the key advantages of this model over conventional statistical approaches.

## Study limitation

The most important limitation of the present study is the missing data, incomplete registrations, and the lack of recorded information in patients' medical records in the registry, which excluded 51 patients from the analysis, although this had very little impact on our findings. Another point that limits the application of flexible cured models is related to classified variables, meaning that the model cannot be fitted for small volumes of variables in categories where the Kaplan-Meier curve remains flat. This issue may diminish with further follow-up of patients and an adequate sample size. Despite attempts to control for confounding variables (e.g., age, sex), residual confounding may still exist. For instance, unmeasured factors like patient adherence to cure or lifestyle choices could influence survival outcomes but were not accounted for in our analysis.

## Conclusion

This study highlights the challenges posed by censoring in survival analysis, particularly when distinguishing between susceptible and non-susceptible individuals. By employing a flexible parametric cure model, a type of non-mixture cure model, our analysis effectively addresses the complexities of survival data in dialysis patients. Notably, we identified 21 predictive variables, with age, calcium, LDL, ESR, and SBP having a significant impact on PD failure. These findings suggest that targeted interventions focusing on these variables could help prevent PD failure.

## Acknowledgments

This study is part of the Master's thesis of Maryam Karimi Ghahfarokhi (No. 8554) in Biostatistics at the Department of Epidemiology and Biostatistics, School of Public Health, Tehran University of Medical Sciences.

## Author contributions

**Conceptualization:** Maryam Karimi Ghahfarokhi, Mostafa Hosseini, Mehdi Yaseri.

**Data curation:** Maryam Karimi Ghahfarokhi.

**Formal analysis:** Maryam Karimi Ghahfarokhi.

**Investigation:** Maryam Karimi Ghahfarokhi.

**Methodology:** Maryam Karimi Ghahfarokhi, Mostafa Hosseini, Mehdi Yaseri.

**Project administration:** Maryam Karimi Ghahfarokhi, Mostafa Hosseini, Mehdi Yaseri.

**Resources:** Maryam Karimi Ghahfarokhi.

**Software:** Maryam Karimi Ghahfarokhi.

**Supervision:** Mostafa Hosseini, Mehdi Yaseri.

**Validation:** Mehdi Yaseri.

**Writing – original draft:** Maryam Karimi Ghahfarokhi.

**Writing – review & editing:** Maryam Karimi Ghahfarokhi, Fatemeh Masaebi.

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
