## [Decision Letter · Decision Letter 0]

PONE-D-24-53137Application of flexible parametric cure model in determination of effective factors on the technique of survival peritoneal dialysis patients in adults of IranPLOS ONE

Dear Dr. Yaseri,

Thank you for submitting your manuscript to PLOS ONE. After careful consideration, we feel that it has merit but does not fully meet PLOS ONE’s publication criteria as it currently stands. Therefore, we invite you to submit a revised version of the manuscript that addresses the points raised during the review process.

We look forward to receiving your revised manuscript.

Kind regards,

Wisit Kaewput, MD

Academic Editor

PLOS ONE

3. In the online submission form, you indicated that your data is available only on request from a third party. Please note that your Data Availability Statement is currently missing the name of the third party contact or institution / contact details for the third party, such as an email address or a link to where data requests can be made. Please update your statement with the missing information.

Reviewers' comments:

Reviewer's Responses to Questions

**Comments to the Author**

1. Is the manuscript technically sound, and do the data support the conclusions?

Reviewer #1: Partly

Reviewer #2: Partly

Reviewer #3: Yes

Reviewer #4: Partly

2. Has the statistical analysis been performed appropriately and rigorously? 

Reviewer #1: Yes

Reviewer #2: I Don't Know

Reviewer #3: Yes

Reviewer #4: No

3. Have the authors made all data underlying the findings in their manuscript fully available?

Reviewer #1: No

Reviewer #2: No

Reviewer #3: Yes

Reviewer #4: No

4. Is the manuscript presented in an intelligible fashion and written in standard English?

Reviewer #1: Yes

Reviewer #2: No

Reviewer #3: Yes

Reviewer #4: No

5. Review Comments to the Author

Reviewer #1: Thank you for giving me an opportunity to review this manuscript titled "Application of flexible parametric cure model in determination of effective factors on the technique of survival peritoneal dialysis patients in adults of Iran". Below is my feedback and a few comments/questions for the authors to incorporate into the manuscript.

Minor comments on the abstract:

1. The term odds ratios (OR) needs to be changed to hazard ratios (HR) in the result section since a Cox-Proportional Hazards model was utilized.

2. ESR typically stands for Erythrocyte Sedimentation Rate. You have used the short form for Inflammatory Index Rate. Please clarify.

Methods/Results:

1. What was the pharmacoepidemiologic study design employed for this study? (Cohort, Case-control, nested case-control etc.)

2. How did you define the baseline (pre-index) and follow-up (post-index) periods. What was the index event? Was it receipt of first PD in the registry?

3. The event of interest, discontinuation of PD has two types of patients; (1) patients who died and (2) patients who discontinued PD. Are both these events considered “uncured” patients in your flexible parametric model? If yes, it would impact your “uncured or susceptible” estimates since patients who discontinued PD may not necessarily die and could very well shift to hemodialysis, transplant or other options. In this case, they would get censored which is not appropriate to your grouping of interest since you want them in the “uncured or susceptible” group. As a result, a disproportionately higher number of patients would get censored as seen in the results section. Is there a better way to define the outcome? How did you define discontinuation of PD as an outcome in the database?

4. Please rearrange the Maller and Zhou test explanation from the results to the methods section to test the following assumptions: (1) presence of patients with long-term survival; (2) sufficient duration of the study period.

5. Is a backward variable selection model the most appropriate approach for identifying baseline risk factors? A backward selection model may tend to eliminate potential risk factors that are clinically meaningful but that aren’t statistically significant in your case at p<0.2. In such cases it would be better to rely on expert opinion. Furthermore, some of the clinically meaningful variables that get excluded as a result of a lack of statistical significance may have a potential interaction with other variables. Did the study account for any covariate interactions?

6. Please specify additional details regarding what type of a parametric cure model was utilized for this study? Were restricted cubic splines used? Did the model account for excess hazard in addition to expected hazard to calculate the overall hazard function? In the flexible parametric cure model did the authors model log hazard or log cumulative hazard?

7. Results lines 157 – 174 need to also be provided in the form of equations. You can include this as supplementary material. The paragraph would be easier for the reader to follow with a set of guiding equations.

8. Should the estimate on line 168 be 0.940 instead of 0.490? Since 0.490 is obtained previously from 1-KM estimator whereas 0.940 has been obtained from exponential distribution. Please verify.

9. Figure 2 indicates that the flexible parametric model curve is almost similar to the standard KM curve but contrastingly different than the other two mixture models. How is using a flexible cure parametric model better than the standard semiparametric KM model for your study?

10. Please provide Table 1 to show what was the sample distribution of patients in the cured and non-cured group across different baseline demographic and clinical characteristics.

Figures:

1. In Fig 1, please specify the unit of time on x-axis. I assume it is years.

Reviewer #2: The study, is of significant value for both clinical practice and research. Peritoneal dialysis (PD) is an important treatment for kidney diseases in the final stage, and the understanding of the factors that influence the survival of the patient can lead to improved care, personalized interventions and better long-term results. The research aims to identify variables, reduce complications and optimize the treatment protocols. The use of flexible parametric cure models provides expanded statistical findings, in particular useful to examine survival in scenarios in which some patients achieve long -term stability. Compared to conventional methods for survival analysis, these models explain time -varying effects and variations of the patient results.

This study also has a broader impact on health policy and resource allocation, which contributes clinical training and treatment strategies. In addition, it contributes to the academic area of survival analysis and promotes the introduction of these advanced methods in other areas of medical research.

However, in its current form, the study has a number of weaknesses that need to be addressed.

1. In the introduction, particularly, in lines 81 to 87, a clearer reason for the selection of the flexible parametric cure model would strengthen the argument against conventional survival models. While the benefits of the model, such as its ability to account for time-varying effects and long-term survival, are noted, elaborating on why it is particularly suited to this study's objectives would add depth. The provision of specific comparisons with conventional models such as Kaplan-Meier or Cox proportional hazards could highlight its advantages more effectively. The inclusion of newer or globally relevant studies could offer a stronger basis and a stronger context for the research.

2. In the materials and methods section, particularly, in lines 126 to 140, the statistical methodology is adequately described, but would benefit from a discussion on model diagnostics and validation techniques applied and their affirmation of the statistical model. While the flexible parametric cure model is introduced and its benefits outlined, explaining how the model’s assumptions were tested and its fit evaluated would provide additional rigor to the analysis. If the author's feel that this makes the Methods section bulky or too technical, a supplementary paper on these aspects could be added to the documents submitted for review.

3. The discussion would benefit from a more detailed comparison with similar studies, particularly in exploring how the findings from this model align with or differ from previous research, employing different models. Highlighting areas of divergence or agreement can provide important context for understanding the contribution of this study to the broader field. Such comparisons would also strengthen the argument for using the flexible parametric cure model by demonstrating its unique insights or confirming its consistency with established findings. Additionally, the potential clinical implications of identifying significant risk factors for peritoneal dialysis (PD) failure could be expanded. A deeper exploration of how these findings can influence patient management, treatment protocols, or policy decisions would enhance the study’s practical relevance. Discussing actionable steps based on the results, such as targeted interventions for high-risk patients or strategies to improve PD outcomes, would further highlight the study's value for clinicians and healthcare systems.

4. On references, given that the study is promoting the use of a relatively recent statistical method, some of the references may be too old to be used in this investigation. For example, for an updated perspective on survival analysis techniques, relevant to this study, refer to the study by Crowther and Lambert (2023), titled “Evaluation of Flexible Parametric Relative Survival Approaches for Enforcing Long-Term Constraints When Extrapolating All-Cause Survival” which evaluates flexible parametric relative survival approaches, instead of Kleinbaum and Klein, 1996. Another example is to use the paper titled, “A Flexible Parametric Modelling Framework for Survival Analysis” by Burke et al. (2019), for recent developments in handling censored and truncated data, which presents a flexible parametric modeling framework encompassing various survival distributions, instead of Klein and Moeschberger, 2006.

5. In its current form, the study's language and phrasing are occasionally unclear, which could make it hard for readers to fully grasp its intended meaning. A comprehensive proofreading of the manuscript is advised to enhance readability and overall quality. Additionally, the writing could be improved by employing a professional language editing service to make sure it is accurate and clear.

Reviewer #3: This study makes a valuable contribution to the field of nephrology. There is acknowledgement of the study limitations in the form of missing data. The authors have performed a comprehensive variable analysis. Would recommend that the authors add more clarity to their complex statistical methods used. Overall, a high-quality study.

Reviewer #4: The authors present an article titled “Application of Flexible Parametric Cure Model in Determining Effective Factors on the Survival of Peritoneal Dialysis Patients in Adults of Iran.”

The following concerns should be addressed to enhance clarity, coherence, and statistical rigor:

1. Clarity of Study Objectives

The scientific rationale is unclear, making it difficult to understand the study’s objectives.

(a) Definition of Peritoneal Dialysis (PD): PD is a treatment for kidney failure that replaces kidney function but does not cure the disease. This should be clearly stated.

(b) Terminology – “Cured” vs. “Non-Cured” Patients: The study refers to “non-cured dialysis patients” and a “cured fraction.” However, dialysis-dependent patients (ESRD) are not considered “curable” unless they receive a kidney transplant or recover from acute kidney injury (AKI). The data does not clarify which category of patients is being analyzed.

(c) Clarification of PD Failure: The statement “PD is considered unsuccessful when kidney function is not adequately replaced” needs refinement. PD does not restore kidney function but only manages the disease.

2. Justification for Study Design

A retrospective cohort study is typically used for rare exposures/events. However, kidney failure (ESRD) is a common condition, especially among those with diabetes, hypertension, and CKD. What is the justification for using this study design?

3. Model Assumptions & Statistical Testing

(a) How were the assumptions of the models verified?

(b) What specific statistical tests were performed?

(c) Where are the results of these tests (main manuscript or supplementary materials)?

4. Interpretation of Results

(a) Line 49-50: The study reports a cure fraction of 50% (95% CI: 0.47-0.53). How does this percentage relate to the confidence interval?

(b) Fig. 2: Both the Kaplan-Meier model and flexible parametric cure model provide similar fits. Why is the flexible model chosen? Table 1 does not present AIC results for Kaplan-Meier—any justification?

5. Inconsistencies in Data Reporting

(a) Line 150: Data collected until February 2018

(b) Line 40: Study conducted between 1995 and 2017

(c) Line 104: Data recorded from 1995 to February 2018

These inconsistencies should be clarified.

6. Kaplan-Meier Analysis

(a) Line 155: The reference to Kaplan-Meier survival plots (77%, 64%) is unclear—please specify the exact figure or table.

(b) Line 151-153: The sentence about Kaplan-Meier survival curves with 95% confidence bands should be checked for correctness.

7. Introduction, Methods, Discussion & Conclusion

(a) The introduction is too brief and lacks sufficient background.

(b) The model assumptions and statistical expressions must be explicitly stated and clearly explained.

(c) The conclusion is weak and does not effectively summarize key findings.

(d) The discussion should be better aligned with previous studies to ensure coherence.

8. Additional Issues

(a) Line 223: What is the estimated failure rate of PD in Iranian adults?

(b) Line 226-227: The statement “After adjusting for confounding factors…” should be justified, as this is a retrospective cohort study, not an experimental study.

9. Language & Writing Style

The manuscript contains grammatical and language errors that need improvement for clarity and readability.

6. PLOS authors have the option to publish the peer review history of their article (what does this mean? ). If published, this will include your full peer review and any attached files.

**Do you want your identity to be public for this peer review?** For information about this choice, including consent withdrawal, please see our Privacy Policy .

Reviewer #1: No

Reviewer #2: No

Reviewer #3: **Yes: ** Sandesh Murali

Reviewer #4: No

---

## [Author Response · Author response to Decision Letter 1]

9 Apr 2025

Dear [Editor/Referee],

Thank you for reviewing our manuscript. We have carefully considered the referee's comments and have made the necessary revisions as requested. Both the detailed responses to the comments and the revised manuscript have been uploaded for your review.

Below are two examples of our responses to the comments:

Dear [Editor/Referee],

Thank you for reviewing our manuscript. We have carefully considered the referee's comments and have made the necessary revisions as requested. We have uploaded both the detailed responses to the comments and the revised manuscript for your review. If any further modifications are required, please let us know.

Below are some example our responses to each of the comments:

for example:

Reviewer 1:

Comment 1: The term odds ratios (OR) needs to be changed to hazard ratios (HR) in the result section since a Cox-Proportional Hazards model was utilized.

Response: Edited

Comment 2: ESR typically stands for Erythrocyte Sedimentation Rate. You have used the short form for Inflammatory Index Rate. Please clarify.

Response: Edited

Methods/Results:

1. What was the pharmacoepidemiologic study design employed for this study? (Cohort, Case-control, nested case-control etc.).

Response: retrospective (historical) cohort

2. How did you define the baseline (pre-index) and follow-up (post-index) periods? What was the index event? Was it receipt of first PD in the registry?

Response: In our study, the index event was defined as the initiation of peritoneal dialysis (PD), which corresponds to the first recorded PD treatment in the Iranian PD registry. This is consistent with standard practices in survival analysis of dialysis patients, where the start of dialysis is typically used as the index event.

Baseline (pre-index) period: The baseline period was defined as the time prior to the initiation of PD. During this period, demographic, clinical, and laboratory data were collected from the registry. These data were used to characterize the patients at the start of PD and to assess potential predictors of survival. For example, laboratory values such as hemoglobin, creatinine, and albumin were recorded as the most recent measurements prior to PD initiation.

Follow-up (post-index) period: The follow-up period began at the initiation of PD (index event) and continued until the occurrence of the event of interest (discontinuation of PD due to technique failure or death) or censoring (e.g., switching to hemodialysis or being placed on the kidney transplant emergency list). Patients were followed until the end of the study period (February 2018) or until they were lost to follow-up.

3. The event of interest, discontinuation of PD has two types of patients; (1) patients who died and (2) patients who discontinued PD. Are both these events considered “uncured” patients in your flexible parametric model? If yes, it would impact your “uncured or susceptible” estimates since patients who discontinued PD may not necessarily die and could very well shift to hemodialysis, transplant or other options. In this case, they would get censored which is not appropriate to your grouping of interest since you want them in the “uncured or susceptible” group. As a result, a disproportionately higher number of patients would get censored as seen in the results section. Is there a better way to define the outcome? How did you define discontinuation of PD as an outcome in the database?

Response: We acknowledge the reviewer's concern regarding the potential impact of discontinuation of peritoneal dialysis (PD) for reasons other than death on the estimation of the "uncured" or "susceptible" population in the flexible parametric cure model.

To address this concern, we would like to clarify the following:

In our study, we defined the event of interest as the discontinuation of PD, which encompasses two primary scenarios:

Death: Patients who died while on PD were classified as having experienced the event of interest. This definition aligns with the clinical objective of assessing long-term survival in PD patients, as death serves as the most definitive endpoint.

Transition to Other Therapies: Patients who transitioned to hemodialysis or were placed on the kidney transplant emergency list were considered censored. This decision was made to ensure that patients who were still receiving care but had changed modalities were not mistakenly classified as having failed PD.

We recognize that censoring patients who transitioned to hemodialysis or were placed on the transplant list could impact our "uncured" or "susceptible" estimates. However, our rationale for this classification is based on the following:

Clinical Relevance: Transitioning to hemodialysis or being listed for a transplant indicates ongoing medical management and does not necessarily reflect a failure of PD. These patients may still be at risk of adverse outcomes, but they are not classified as having "failed" PD in the traditional sense.

Data Integrity: By censoring these patients, we aimed to maintain the integrity of our survival analysis, focusing on those who experienced the event of interest (death) while on PD.

However, the use of a flexible parametric survival model allowed us to estimate survival rates while accounting for the complexities of our dataset.

Reviewer 2:

Comment 1: In the introduction, particularly, in lines 81 to 87, a clearer reason for the selection of the flexible parametric cure model would strengthen the argument against conventional survival models. While the benefits of the model, such as its ability to account for time-varying effects and long-term survival, are noted, elaborating on why it is particularly suited to this study's objectives would add depth. The provision of specific comparisons with conventional models such as Kaplan-Meier or Cox proportional hazards could highlight its advantages more effectively. The inclusion of newer or globally relevant studies could offer a stronger basis and a stronger context for the research.

Response:

Thank you for your valuable suggestions. I have revised the manuscript accordingly and made the following changes:

“Historically, studies analyzing CAPD patient survival have predominantly relied on conventional statistical models, such as the log-rank test and the Cox proportional hazards model (6–8). However, these models assume that all individuals in the study are at risk of experiencing the event (death) or being censored. This assumption does not align with PD, as it is a maintenance treatment where a substantial proportion of patients do not experience death during follow-up. For example, while Kaplan-Meier survival estimates provide useful insights, they fail to account for patients who are effectively "cured" of the event of interest, leading to potential biases. Similarly, the Cox proportional hazards model assumes that all individuals remain susceptible to the event throughout the study period, which does not adequately capture the heterogeneity in PD populations.

Reference:

7- Kleinbaum and Klein (1996) were replaced by Crowther and Lambert (2023)

8- Klein and Moeschberger (2006) were replaced by Burke et al. (2019)

A more appropriate approach involves cure models (1-6), which divide the population into two groups: susceptible individuals and those considered cured or immune from the event. Cure models assume that a subset of patients will never experience the event, resulting in a plateau in the survival curve. These models estimate both the proportion of cured individuals and the survival function of those who remain susceptible, while also examining the factors influencing both outcomes (9). Cure models are broadly classified into two types: mixture models and non-mixture models (10). The reliability of these models depends on various factors, including the duration of follow-up and the selection of an appropriate survival distribution (e.g., exponential, Weibull, or logistic) for susceptible individuals (11).”

Here are some newer and globally relevant studies that could provide a stronger foundation and context for our research on cure models in survival analysis. These studies will be incorporated into the article.

1. Amico, M., & Van Keilegom, I. (2018). Cure models in survival analysis. Annual Review of Statistics and Its Application, 5, 311–342. https://doi.org/10.1146/annurev-statistics-031017-100101

2. Legrand, C., & Bertrand, A. (2019). Cure models in cancer clinical trials. In Textbook of Clinical Trials in Oncology (pp. 123–145). CRC Press. https://doi.org/10.

3. Grant, T. S., Burns, D., Kiff, C., & Lee, D. (2020). A case study examining the usefulness of cure modelling for the prediction of survival based on data maturity. Pharmacoeconomics, 38(5), 521–532. https://doi.org/10.1007/s40273-020-00895-6

4. Whiting, K., Fei, T., Singer, S., & Qin, L. X. (2024). Cureit: An end-to-end pipeline for implementing mixture cure models with an application to liposarcoma data. JCO Clinical Cancer Informatics, 8, e2300201. https://doi.org/10.1200/CCI.23.00201

5. Maller, R., Resnick, S., Shemehsavar, S., & Zhao, M. (2024). Mixture cure model methodology in survival analysis: Some recent results for the one-sample case. Statistical Surveys, 18, 1–25. https://doi.org/10.1214/23-SS123

6. Filleron, T., Bachelier, M., Mazieres, J., Pérol, M., Meyer, N., Martin, E., Mathevet, F., Dauxois, J.-Y., Porcher, R., & Delord, J.-P. (2021). Assessment of treatment effects and long-term benefits in immune checkpoint inhibitor trials using the flexible parametric cure model: A systematic review. JAMA Network Open, 4(12), e2139573. https://doi.org/10.1001/jamanetworkopen.2021.39573

---

## [Decision Letter · Decision Letter 1]

Application of flexible parametric cure model in determination of effective factors on the technique of survival peritoneal dialysis patients in adults of Iran

PONE-D-24-53137R1

Dear Dr. Yaseri,

We’re pleased to inform you that your manuscript has been judged scientifically suitable for publication and will be formally accepted for publication once it meets all outstanding technical requirements.

Kind regards,

Wisit Kaewput, MD

Academic Editor

PLOS ONE

Additional Editor Comments (optional):

Accept as is.

Reviewers' comments:

Reviewer's Responses to Questions

**Comments to the Author**

1. If the authors have adequately addressed your comments raised in a previous round of review and you feel that this manuscript is now acceptable for publication, you may indicate that here to bypass the “Comments to the Author” section, enter your conflict of interest statement in the “Confidential to Editor” section, and submit your "Accept" recommendation.

Reviewer #1: All comments have been addressed

Reviewer #2: All comments have been addressed

Reviewer #4: All comments have been addressed

2. Is the manuscript technically sound, and do the data support the conclusions?

Reviewer #1: Yes

Reviewer #2: (No Response)

Reviewer #4: Yes

3. Has the statistical analysis been performed appropriately and rigorously? 

Reviewer #1: Yes

Reviewer #2: (No Response)

Reviewer #4: Yes

4. Have the authors made all data underlying the findings in their manuscript fully available?

Reviewer #1: Yes

Reviewer #2: (No Response)

Reviewer #4: Yes

5. Is the manuscript presented in an intelligible fashion and written in standard English?

Reviewer #1: Yes

Reviewer #2: (No Response)

Reviewer #4: Yes

6. Review Comments to the Author

Reviewer #1: (No Response)

Reviewer #2: (No Response)

Reviewer #4: The authors have made commendable efforts to improve the manuscript, and the revisions are appreciated. However, can the authors consider presenting the fitted results from various models visually—using different colour codes to overlay the model fits on the data. This will add substantial value and better support the novelty of your contribution.

7. PLOS authors have the option to publish the peer review history of their article (what does this mean? ). If published, this will include your full peer review and any attached files.

**Do you want your identity to be public for this peer review?** For information about this choice, including consent withdrawal, please see our Privacy Policy .

Reviewer #1: No

Reviewer #2: No

Reviewer #4: No

---

## [Editor Report · Acceptance letter]

PONE-D-24-53137R1

PLOS ONE

Dear Dr. Yaseri,

I'm pleased to inform you that your manuscript has been deemed suitable for publication in PLOS ONE. Congratulations! Your manuscript is now being handed over to our production team.

Kind regards,

on behalf of

Dr. Wisit Kaewput

Academic Editor

PLOS ONE